# Inulin, Choline and Silymarin in the Treatment of Irritable Bowel Syndrome with Constipation—Randomized Case-Control Study

**DOI:** 10.3390/jcm11082248

**Published:** 2022-04-17

**Authors:** Oana-Bogdana Bărboi, Ioan Chirilă, Irina Ciortescu, Carmen Anton, Vasile-Liviu Drug

**Affiliations:** 1Department of Gastroenterology, ‘Grigore T. Popa’ University of Medicine and Pharmacy, 700115 Iasi, Romania; oany_leo@yahoo.com (O.-B.B.); carmen_ro2008@yahoo.com (C.A.); vasidrug@email.com (V.-L.D.); 2Institute of Gastroenterology and Hepatology, ‘Saint Spiridon’ Hospital, 700111 Iasi, Romania; 3Environmental Health Department, National Institute of Public Health—RCoPH, 700465 Iasi, Romania; chirilaioan@gmail.com

**Keywords:** irritable bowel syndrome, constipation, inulin, choline, silymarin

## Abstract

(1) Background: Irritable bowel syndrome (IBS) is a common disease, with multiple pathophysiological mechanisms involved. A single treatment for all the patients with IBS is not possible. Prebiotics may have a beneficial effect on IBS patients with constipation. (2) Methods: A randomized cross-over case-control study was conducted, including patients with IBS and constipation (IBS-C), who were randomized into two groups receiving a specific constipation diet with or without a food supplement containing inulin, choline and silymarin (Stoptoxin^®^, Fiterman Pharma, Iasi, Romania). Patients were evaluated at baseline, after four and eight weeks, using a questionnaire to assess IBS symptoms. (3) Results: 51 IBS-C patients were included, of which 47 patients finished the trial (33 women, mean age 52.82 years). Adding Stoptoxin^®^ to a diet for constipation brought extra benefits. Abdominal pain severity improved by 68.3% after the diet and Stoptoxin^®^ (*p* = 0.004) and abdominal bloating severity parameter improved by 34.8% (*p* = 0.040). The stool number per week and the stool consistency according to the Bristol scale were improved, but without statistical significance between groups (*p* > 0.05). (4) Conclusions: The combination of inulin, choline and silymarin associated with a specific-constipation diet had obvious clinical beneficial effects on IBS-C patients in terms of bowel movement, abdominal pain and bloating.

## 1. Introduction

Irritable bowel syndrome (IBS) is one of the most prevalent conditions of the gut–brain interaction disorders. It continues to represent a significant burden to the health care system and to individuals. It may also substantially alter the patient’s quality of life. The ROME IV consensus provided the diagnosis criteria for IBS and classified IBS into four subgroups, using the Bristol scale: IBS with constipation (IBS-C), IBS with diarrhea (IBS-D), the mixed form (IBS-M) and the undefined group (IBS-U) [1]. Due to both clinical heterogeneity and different pathophysiological pathways implied in developing IBS, a universal approach is not feasible. In recent decades, a multitude of drugs have been developed to treat IBS, being directed towards symptoms. However, treating IBS patients may be quite difficult, as patients with the same clinical manifestations may respond differently to the same medication. Probiotics and prebiotics, which interfere with gut microbiota, may have a beneficial effect on IBS symptoms [2].

Inulin is a nondigestible oligosaccharide which seems to regulate the intestinal transit and the stool consistency and frequency, and to modulate the immune response. Moreover, inulin is considered a prebiotic, characterized as ‘functional fiber’, with several pharmaceutical and food applications [3,4]. The efficacy of treatment with inulin of IBS patients is reflected by some studies that highlight the importance of consuming inulin as part of a regular daily diet, especially in patients with constipation [5,6,7].

Limited studies on IBS-C patients have silymarin and choline, which seems to stimulate intestinal peristalsis, as they stimulate the biliary secretion [8,9]. There is a dose-dependent relationship between sylimarin and the secretory liver function stimulation [8]. Choline is a bowel motility promoter which is appropriate for use for constipation [9].

The aim of this study was to assess if an inulin combined choline and silymarin may exert a beneficial effect on IBS-C patients in terms of symptomatology and quality of life and to compare it with a specific daily IBS-C diet.

## 2. Materials and Methods

### 2.1. Study Protocol

A randomized cross-over study was performed in a tertiary center in the north-east of Romania. Fifty-one adult patients diagnosed with IBS-C according to the ROME IV criteria were included in the study. They had no history of pre/probiotic or laxatives consumption in the last 10 days. Pregnant or lactating women were excluded from the study.

The closed envelope method was used for randomization. The patients were assigned into two groups: Group A received a constipation-specific diet (based on fibers, dairy, fruits) (Appendix A) and group B received both the diet and a product containing 5000 mg inulin extracted from *Cichorium inthybus* root, 37 mg choline and 40 mg silymarin (Stoptoxin^®^, Fiterman Pharma, Iasi, Romania), as follows: One sachet per day in the first week and two sachets per day over the next three weeks. After 28 days, the two groups were swapped. All the patients were evaluated initially (visit 1), at day 28 (visit 2) and at day 57 (visit 3). The following parameters were assessed: Stool habits, stool characteristics (Bristol Scale) and symptom severity (abdominal pain, frequency of abdominal pain, bloating severity, patient satisfaction in relation with stool frequency and the impact of IBS on daily activity) as appreciated by the patients for the last 10 days. The study design is presented in Figure 1.

For each of the IBS symptom parameters, a score between 0 and 100 was given (Appendix B). A total score was obtained by the sum of the five individual scores, between 0 and 500. A higher symptoms score indicated more severe symptoms. Using this score, the patients were grouped into the following severity SII symptoms groups:-Mild severity—total score less than 175;-Moderate severity—total score between 175 and 300;-High severity—total score over 300.

A significant clinical improvement was considered to be a reduction in the overall severity score of IBS symptoms of at least 50 points.

In order not to interfere with the Stoptoxin^®^ effect, patients avoided the use of other products with known laxative effects. All the medication or dietary supplements used by the patients during the study were recorded in the patients’ files.

The study was registered on ClinicalTrials.gov with the identifier (NCT number) NCT03174561.

### 2.2. Statistical Analysis

Statistical application MS Excel was used and the *t*-Test: Paired Two Sample for Means was applied. The statistical significance of the differences between the groups was calculated using P (T ≤ t) two tail. A *p*-value less than 0.05 was considered statistically significant.

### 2.3. Ethical Aspects

The study was conducted in accordance with the Declaration of Helsinki. The study protocol was approved by the Ethics Committee of the University of Medicine and Pharmacy “Grigore T. Popa” Iasi and of the“Saint Spiridon” Hospital Iasi, Romania.

## 3. Results

The study included 51 IBS-C patients, 34 female and 17 male, with a mean age of 53.97 years (22–86 years). The mean of patients’ stool frequency per week was 3.83 and the mean of patients’ stool consistency according to the Bristol scale was 2.7. Of 51 subjects enrolled in the study, 28 patients were randomized in group A (starting with diet) and 23 patients in group B (starting with diet and the food supplement). The two groups were similar in terms of demographic features and initial IBS symptoms (*p* > 0.05, *t*-Test: Two Sample Assuming Equal Variances).

At the initial evaluation, we found that three patients (5.89%) had mild symptoms (total symptoms score < 175), 26 patients (50.98%) had moderate symptoms (total symptoms score between 175 and 300) and 22 patients (43.13%) had severe IBS symptoms (total symptoms score > 300) (Figure 2).

From the 51 patients, only 47 patients presented at all three visits: 33 female and 14 men, mean age 52.82 years (22–77 years). The mean frequency of stools per week for these patients was 3.86 and the mean consistency of stools according to the Bristol scale was 2.5. The two groups were similar in terms of demographic features and initial IBS symptoms (*p* > 0.05) (Table 1). Group A included 24 patients (16 women, mean age 52.39 years) and group B included 23 patients (17 women, 53.27 years).

After the first 28 days, the total IBS symptoms score significantly decreased for group B1 patients compared to group A1 patients (148.7 vs. 117.08, *p* < 0.001) (Figure 3). Analyzing the evolution of each parameter, a statistically significant reduction of scores was only recorded for the “IBS influence on patient’s daily activity” parameter (−36.96 points; *p* < 0.05). There was also a reduction of the other parameters’ scores, but without statistical significance (Figure 4).

Regarding the stools frequency and consistency, there was no statistically significant difference between groups after the first four weeks of treatment (Figure 5).

After 56 days, there was also a statistically significant improvement of total IBS symptom scores for the patients who received diet-C and Stoptoxin^®^ versus the patients who continued with diet-C (*p* < 0.01) (Figure 6). Of all assessed parameters, for abdominal pain severity and frequency only, a statistically significant reduction of scores was recorded (*p* < 0.05) (Figure 7).

At visit 3, there was also no statistically significant difference between the two groups regarding consistency of patient’s stools, but the number of stools per week significantly improved (Figure 8).

Adding Stoptoxin^®^ to diet-C brought extra benefits, the clinical improvement being reflected by total symptoms score, which was higher with 30.1% (*p* = 0.034). In the case of the abdominal pain severity parameter, the improvement after diet and Stoptoxin^®^ was greater, with up to 68.3% (*p* = 0.004), and in the case of bloating severity, the improvement was higher, with up to 34.8% (*p* = 0.040). The stools number per week and the stools consistency according to Bristol scale were improved, but without statistical significance between groups (*p* > 0.05) (Table 2).

## 4. Discussion

IBS is one of the most common digestive disorders with multifactorial pathophysiological underlying mechanisms and heterogeneous clinical presentation. The cardinal IBS manifestation is chronic abdominal pain related to defecation, with either changes in stool frequency or form. Symptoms are often severe, with a negative impact on individuals’ daily lives [10]. IBS’s pathogenesis consists of the interaction of genetic, environmental and psychosocial factors [11]. That is the reason why a “one-size-fits-all” approach is not an option.

The prevalence of IBS patients is higher among women, as the most extensive global epidemiological study recently showed [12]. Sperber et al. [12] demonstrated that women have IBS-C more often than IBS-D. IBS affects people of all ages, especially with a range between 30 and 50 years [11]. In our study, most of the patients included were women of middle age who showed moderate or severe IBS-C symptoms.

We assessed the potential beneficial effects of a combination of three substances—inulin, choline and silymarin—along with a specific constipation diet on patients diagnosed with IBS-C, according to the ROME IV criteria.

Choline is an essential nutrient whose deficiency is involved in cardiovascular diseases, irritable and inflammatory bowel diseases, non-alcoholic fatty liver disease and chronic kidney disease, as choline-deficient diets may induce gut microbiota alterations [13].

Potential beneficial effects of silymarin on IBS-C patients are linked to the stimulation of intestinal peristalsis, as silymarin stimulates biliary secretion [8].

Inulin acts as a prebiotic with effects that are linked to the regulation of bowel transit, stool frequency and consistency, being successfully used both for healthy individuals and IBS-C patients [14,15,16]. Prebiotic supplements may modulate gut bacteria and reduce symptoms of IBS-C when taken in low doses [17,18]. The European Food Safety Authority reported that around 12 g of inulin per day is necessary in order to maintain a regular bowel transit, without any side effects [19]. Hond et al. [20] demonstrated a significantly higher stool frequency compared to placebo at an inulin intake of 15 g per day. The inulin intake is also effective in improving stool consistency according to Bristol scale, as it reduces stools’ hardness, as well as intestinal transit time [15]. Compared to placebo, inulin increased patients’ satisfaction with constipation [7,21,22,23].

The present study strengthens the effect of inulin on IBS-C patients reported in the trials mentioned above. The diet based on fibers is efficient for IBS-C patients. However, adding the medicine containing inulin, choline and silymarin proved to be a much more attractive therapeutic option for reduction of IBS symptoms, especially abdominal pain severity and frequency, abdominal bloating, and also for improvement of patients’ quality of life, without any side effects. Our study did not demonstrate an improvement of bowel habits in terms of number of stools per week and stool consistency when comparing diet-C and Stoptoxin^®^ with diet-C alone.

Our data collected at visit 3 suggested that stopping Stoptoxin^®^ and continuing only with diet-C leads to losing the beneficial clinical effects obtained after 28 days of treatment. Therefore, we can speculate that periodic short courses of Stoptoxin^®^ and diet-C may be useful for IBS-C patients, but we did not evaluate this in the present study.

Analyzing each IBS symptom parameter assessed in our study after 28 and 56 days, the results were inconsistent, which in fact we were actually expecting due to the study design. We found that after four weeks of an intake of 10 g per day of inulin contained in Stoptoxin^®^ along with the diet-C, an improvement of patients’ quality of life was identified. Instead, an improvement of both bowel habits (reflected by a higher number of stools per week) and abdominal pain (frequency and severity parameters) was found in patients who received first only the diet and then Stoptoxin^®^ and the diet.

Our study’s limitations are caused by the small sample of patients assessed from each group; moreover, four patients did not complete the study. However, the low drop-out number of patients did not significantly influence the statistical data.

## 5. Conclusions

The combination of inulin, choline and silymarin associated with the constipation-specific diet had obvious clinical beneficial effects on IBS-C patients in terms of bowel movement, abdominal pain and bloating. The lack of side effects if it is consumed in the recommended dosage makes Stoptoxin^®^ and diet-C a reasonable approach for IBS-C patients.

## Figures and Tables

**Figure 1 jcm-11-02248-f001:**
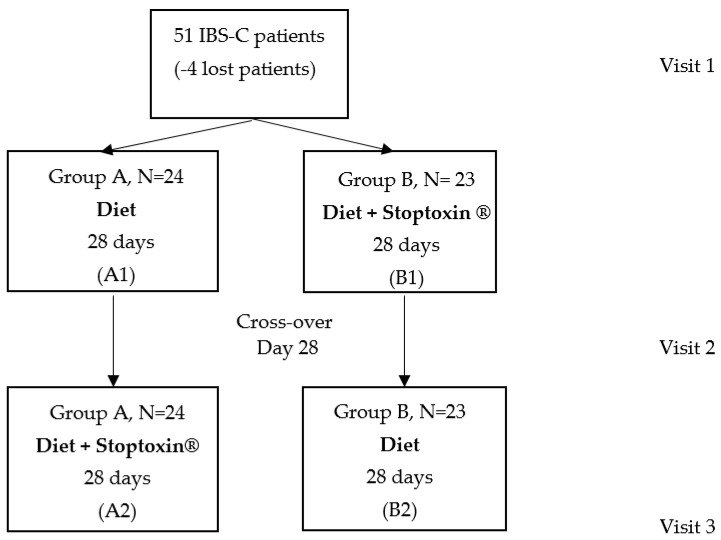
Study design.

**Figure 2 jcm-11-02248-f002:**
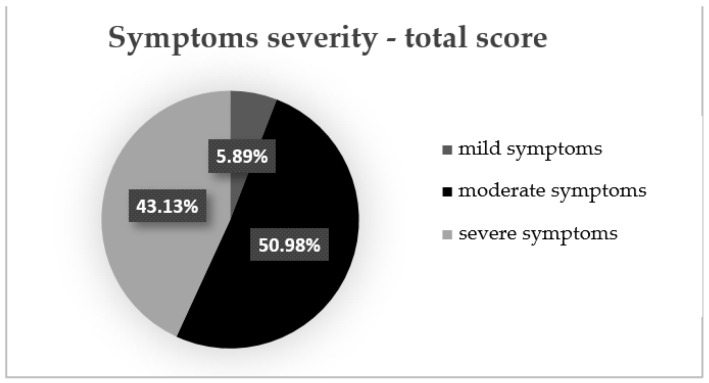
Symptoms severity—total symptoms score.

**Figure 3 jcm-11-02248-f003:**
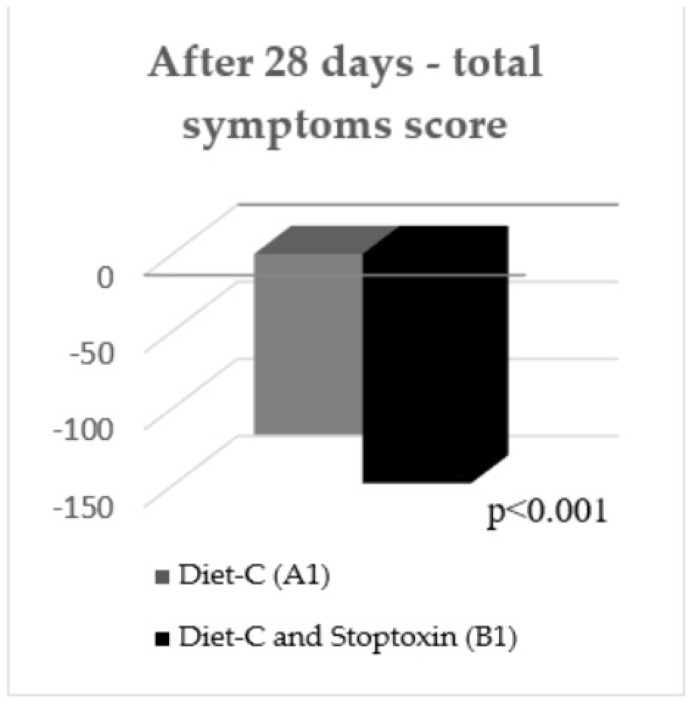
IBS total symptoms score after 28 days.

**Figure 4 jcm-11-02248-f004:**
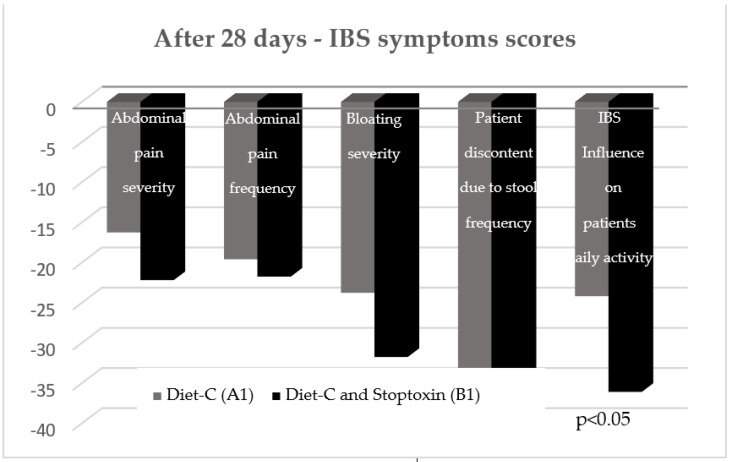
IBS symptoms scores after 28 days.

**Figure 5 jcm-11-02248-f005:**
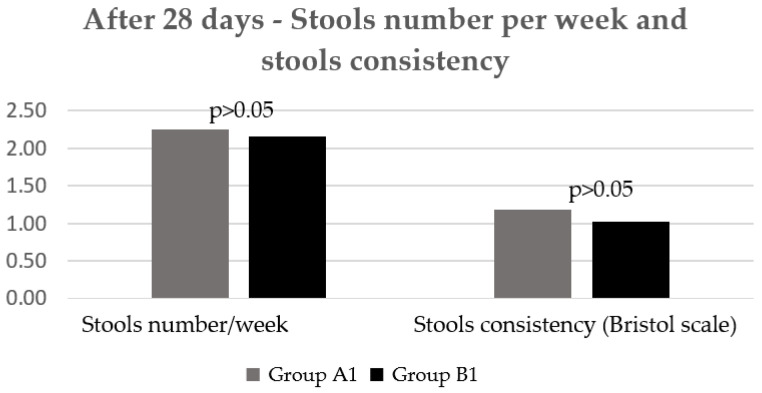
Number/week and consistency of stools after 28 days.

**Figure 6 jcm-11-02248-f006:**
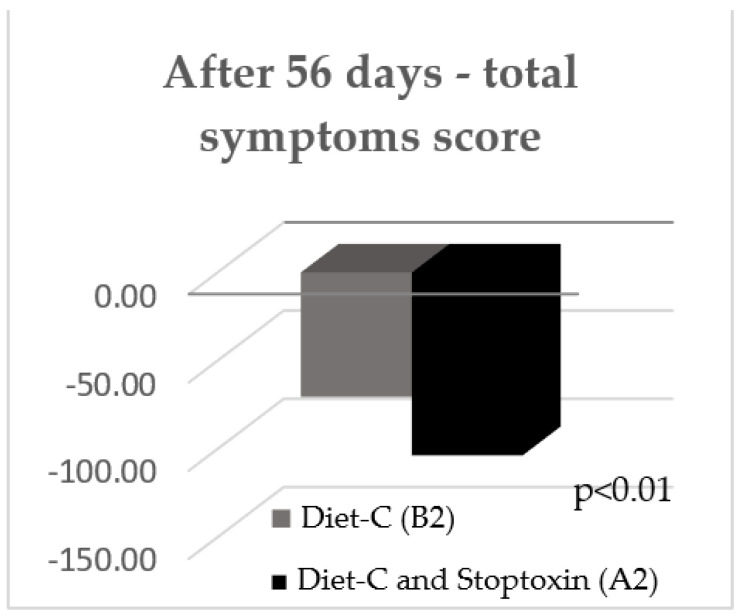
IBS symptom scores after 56 days.

**Figure 7 jcm-11-02248-f007:**
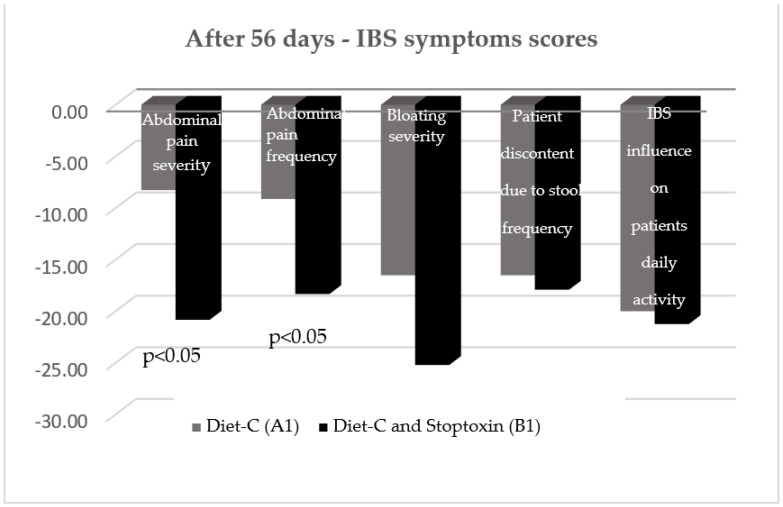
IBS symptom scores after 56 days.

**Figure 8 jcm-11-02248-f008:**
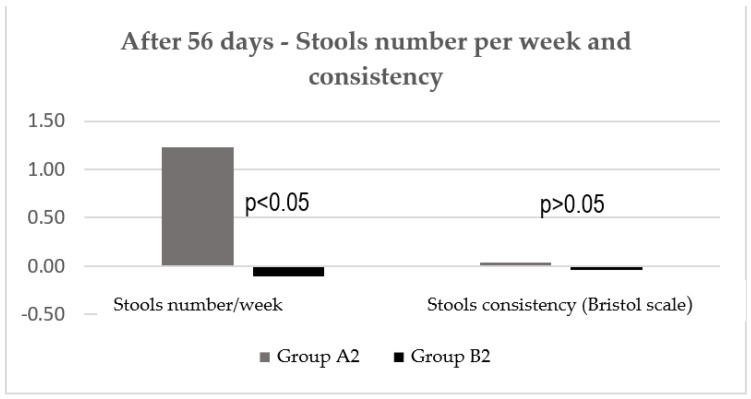
Stools number/week and stools consistency after 56 days.

**Table 1 jcm-11-02248-t001:** IBS parameters of the 47 patients at initial evaluation.

	Number of Stools/Week	Stool Cosistency (Bristol Scale)	Abdominal Pain Severity	Abdominal Pain Frequency	Bloating Severity	Patient Discontent Due to Stool Frequency	IBS Influence on Patient’s Daily Activity	Total Symptoms Score
Mean	3.80	2.49	45.5	46.2	65.7	67.7	66.2	291.3
SD	1.94	1.16	24.47	25.32	25.93	28.14	23.08	82.68
Mean group A	3.27	2.21	49.17	49.58	65.42	69.17	64.17	297.50
Mean group B	4.36	2.77	41.74	42.61	66.09	66.09	68.26	284.78
*p* value	>0.05	>0.05	>0.05	>0.05	>0.05	>0.05	>0.05	>0.05

**Table 2 jcm-11-02248-t002:** The symptoms improvement with two regimens.

	Number of Stools/Week	Stool Cosistency (Bristol Scale)	Abdominal Pain Severity	Abdominal Pain Frequency	Bloating Severity	Patient Discontent Due to Stool Frequency	IBS Influence on Patients Daily Activity	Total Symptoms Score
Diet (A1 + B2)	1.095	0.58	−12.77	−14.68	−21.06	−25.32	−22.77	−96.60
Stoptoxin^®^ + diet (B1 + A2)	1.68	0.51	−21.49	−20.00	−28.40	−27.23	−28.51	−125.64
% suplimentary of improvement in case of using Stoptoxin^®^ (B1 + A2) vs. (A1 + B2)	53.42	12.07	68.3	36.2	34.8	7.6	25.2	30.1
Statistical significance (*p*)	0.346	0.829	0.004	0.092	0.040	0.678	0.139	0.034

IBS: irritable bowel syndrome; A1: group of IBS patients that received constipation-specific diet in the first 28 days; A2: group of IBS patients that received constipation-specific diet and Stoptoxin® in the first 28 days; A2: group of IBS patients that received constipation-specific diet and Stoptoxin® in the next 28 days; B2: group of IBS patients that received constipation-specific diet in the next 28 days.

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
