# Peer review of "Inulin, Choline and Silymarin in the Treatment of Irritable Bowel Syndrome with Constipation—Randomized Case-Control Study"

_jcm, 2022, doi:10.3390/jcm11082248_

Round 1

Reviewer 1 Report

major revision 

1.The author should explain in the preface why Inulin, choline and silymarin are used. Each of the three drugs should be described briefly

2.Authors should clearly indicate the dosage of choline and silymarin used. Are there any other side effects? Since choline is a very obvious agent of bowel motility, it should be clearly stated whether it is appropriate to use it here or as a positive agent.

3.From the table and graph, it is difficult to know clearly which group used Inulin choline. Use of both drugs should be clearly indicated on the application or form.

Mini  revision 

1.line 126 ,142,  women should modified as female. In the paper,you should modified as female.

2.Table 1 and 2 should be expressed in the tird line,

 3.The authors Should decay what mean T test  A vs B, the data of 2.0153 ?

Reviewer 2 Report

Inulin, choline and silymarin in the treatment of irritable bowel syndrome with constipation – randomized case-control study

This is a small randomized cross-over clinical trial examining the effects of a mixture of inuline, choline and silmaryin on patients with IBS-C. Overall the research question is relevant and interesting but there are some methodological issues.

Major comments

  1. Why have the authors not used an existing and validated IBS symptom questionnaire? Is the one they are using validated?
  2. Was there a wash-out period used? Any data on how long the effect of stoptoxin lasts?
  3. I have some issues with the statistical testing involved. I don’t think a paired t test suffices for the kind of data analyzed. A lineair mixed model seems more appropriate
  4. Why did the authors choose the combination of inuline, choline and silymarin. Especially as the authors mention in the discussion that inuline has a known effect on constipation and transit. What is the added benefit of choline and silymarin – and why is the inuline not used as a comparator?

Minor comments

Introduction

Some typo’s/problems with spelling – I recommend asking a native English speaker to go through the text

Pg 1 line 31 one of the most prevalent conditions

Pg 1 line 37 unique => universal?

Methodology

Figure 1 – reasons for lost to follow-up?

Figure 2 – should be supplemental

Please give some more information regarding the constipation diet and what it involves, how it was explained to the patients

Results

A table with baseline characteristics of the population is missing (age, number of years of IBS, how many lines of treatment they had etc)

Round 2

Reviewer 1 Report

The authors have revised and improved the puzzling part of my annotation, and my problem has been solved.

I suggest the manuscript could be accepted.